

# *Wolbachia* co-infection in a hybrid zone: discovery of horizontal gene transfers from two *Wolbachia* supergroups into an animal genome

Lisa J. Funkhouser-Jones[1], Stephanie R. Sehnert[1], Paloma Martínez-Rodríguez[2,3], Raquel Toribio-Fernández[2], Miguel Pita[2], José L. Bella[2] and Seth R. Bordenstein[1,4]

[1] Department of Biological Sciences, Vanderbilt University, Nashville, TN, United States
[2] Departamento de Biología (Genética), Facultad de Ciencias, Universidad Autónoma de Madrid, Madrid, Spain
[3] INRA, Univ. Nice Sophia Antipolis, CNRS, UMR 1355-7254 Institut Sophia Agrobiotech, Sophia Antipolis, France
[4] Department of Pathology, Microbiology and Immunology, Vanderbilt University, Nashville, TN, United States

Corresponding author
Lisa J. Funkhouser-Jones,
lisa.j.funkhouser@vanderbilt.edu

## ABSTRACT

Hybrid zones and the consequences of hybridization have contributed greatly to our understanding of evolutionary processes. Hybrid zones also provide valuable insight into the dynamics of symbiosis since each subspecies or species brings its unique microbial symbionts, including germline bacteria such as *Wolbachia*, to the hybrid zone. Here, we investigate a natural hybrid zone of two subspecies of the meadow grasshopper *Chorthippus parallelus* in the Pyrenees Mountains. We set out to test whether co-infections of B and F *Wolbachia* in hybrid grasshoppers enabled horizontal transfer of phage WO, similar to the numerous examples of phage WO transfer between A and B *Wolbachia* co-infections. While we found no evidence for transfer between the divergent co-infections, we discovered horizontal transfer of at least three phage WO haplotypes to the grasshopper genome. Subsequent genome sequencing of uninfected grasshoppers uncovered the first evidence for two discrete *Wolbachia* supergroups (B and F) contributing at least 448 kb and 144 kb of DNA, respectively, into the host nuclear genome. Fluorescent *in situ* hybridization verified the presence of *Wolbachia* DNA in *C. parallelus* chromosomes and revealed that some inserts are subspecies-specific while others are present in both subspecies. We discuss our findings in light of symbiont dynamics in an animal hybrid zone.

## INTRODUCTION

Microbial communities of many arthropod species are dominated numerically by heritable bacterial symbionts whose phenotypic effects range from mutualism to parasitism (*Douglas, 2011*). In some cases, millennia of co-evolution have produced obligate, mutualistic relationships in which microbial symbionts make essential amino acids and/or

vitamins to complement the nutritionally incomplete diet of their hosts (*Pais et al., 2008*; *Tamas et al., 2002*; *Van Ham et al., 2003*). In other cases, maternally-transmitted bacteria directly impact arthropod host reproduction by manipulating sex determination, fecundity, and the ratio of infected females (the transmitting-sex) within a population (*LePage & Bordenstein, 2013*). The alphaproteobacterium *Wolbachia* is the most widespread of these reproductive manipulators, infecting an estimated 40–52% of all terrestrial arthropod species (*Weinert et al., 2015*; *Zug & Hammerstein, 2012*). It uses a variety of mechanisms to increase the number of host females in a population including feminization of genetic males, male-killing, parthenogenesis, and cytoplasmic incompatibility (CI), which typically results in embryonic death of offspring produced by an uninfected female mated with an infected male (*Serbus et al., 2008*).

Hybrid zones are excellent model systems for studying the impact of interactions between heritable endosymbionts on animal evolution. For example, *Drosophila recens* and *D. subquinaria* meet in secondary contact in a hybrid zone spanning central Canada where *D. recens* is infected by a *Wolbachia* strain that causes strong CI (∼90% reduction in progeny) when males mate with naturally uninfected *D. subquinaria* females (*Jaenike et al., 2006*; *Shoemaker, Katju & Jaenike, 1999*). In contrast, weak levels of CI in a hybrid zone could promote *Wolbachia* exchange between animal species. Two closely related species of field crickets, *Gryllus firmus* and *G. pennsylvanicus*, hybridize in a north-south zone along the eastern front of the Appalachian Mountains in the United States (*Harrison & Arnold, 1982*). Though each cricket species is predominantly infected with different *Wolbachia* strains, *Wolbachia* is not a primary source of hybrid incompatibility in this system (*Mandel, Ross & Harrison, 2001*). This may partly explain why a significant portion of *G. pennsylvanicus* are infected with both *Wolbachia* strains (*Mandel, Ross & Harrison, 2001*). *Wolbachia* co-infection of the same host can readily facilitate gene exchange and transfer of mobile elements between intracellular bacteria according to the intracellular arena concept (*Bordenstein & Reznikoff, 2005*; *Bordenstein & Wernegreen, 2004*; *Newton & Bordenstein, 2011*). Indeed, we previously showed that the co-infecting *Wolbachia* strains in *G. pennsylvanicus* crickets harbor a nearly identical infection of *Wolbachia's* temperate bacteriophage WO (*Chafee et al., 2010*). Thus, hybrid zones that permit mixing of *Wolbachia* symbionts may in turn enable horizontal gene transfer between the coinfections.

Here, we investigate horizontal gene transfer of bacteriophage WO in a natural hybrid zone of the meadow grasshopper *Chorthippus parallelus*. During the last Ice Age, *C. parallelus* populations on the Iberian Peninsula were geographically isolated from those in continental Europe, resulting in the divergence of Iberian *C. parallelus erythropus* (Cpe) subspecies from the contemporary continental subspecies, *C. parallelus parallelus* (Cpp) (*Shuker et al., 2005a*). Now in secondary contact, hybrids of the two subspecies have interbred for an estimated 9,000 generations along a hybrid zone in the Pyrenees Mountains between France and Spain (*Hewitt, 1993*; *Shuker et al., 2005a*) (Fig. 1). Due to low dispersal rates, all grasshoppers collected from populations in the hybrid zone (i.e., Portalet) are hybrids of the two subspecies, while pure Cpp and Cpe populations

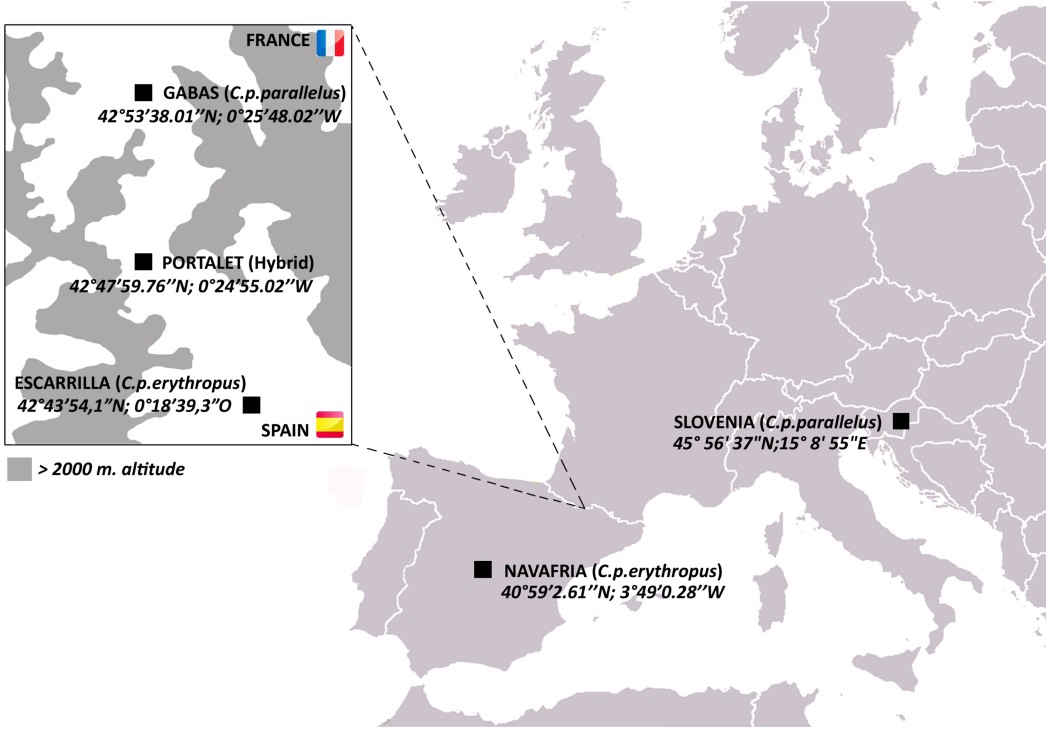

**Figure 1 Map of *C. parallelus* collection sites with their geographical coordinates.** Boxed inset shows the hybrid zone of *C. p. parallelus* and *C. p. erythropus* subspecies in the D'Ossau and Tena valleys of the Pyrenees Mountains between France and Spain.

reside on the edges of the hybrid zone (Gabas for Cpp and Escarrilla for Cpe) (*Bella et al., 2007*; *Hewitt, 1993*; *Shuker et al., 2005a*) (Fig. 1). F1 hybrids produced in laboratory crosses between the subspecies follow Haldane's rule and produce sterile F1 hybrid males, but both hybrid males and females in the field are fertile, possibly due to selection against deleterious allelic combinations that result in hybrid sterility (*Bella, Hewitt & Gosalvez, 1990*; *Shuker et al., 2005b*).

*C. parallelus* subspecies are infected with *Wolbachia* strains from two divergent supergroups: Cpp are primarily infected with B *Wolbachia* while Cpe mostly harbor F *Wolbachia* (*Zabal-Aguirre, Arroyo & Bella, 2010*). In natural hybrid populations, the B and F *Wolbachia* each cause a significant amount of unidirectional CI, reducing embryo viability by approximately 33% and 23%, respectively, in incompatible crosses (*Zabal-Aguirre et al., 2014*). Bidirectional CI is weaker, with a 15% reduction in viable embryos in crosses between F-infected and B-infected grasshoppers (*Zabal-Aguirre et al., 2014*). With these incomplete CI rates permitting the spread of *Wolbachia* strains, the incidence of *Wolbachia* infection is highly variable in the hybrid zone, and individuals collected from a single population are either uninfected, singly-infected with B or F *Wolbachia*, or co-infected by both (*Zabal-Aguirre, Arroyo & Bella, 2010*).

As the temperate bacteriophage WO is well known to transfer between A and B supergroup co-infections in arthropods (*Bordenstein & Wernegreen, 2004*; *Chafee et al., 2010*; *Gavotte et al., 2007*; *Kent et al., 2011*; *Masui et al., 2000*; *Metcalf & Bordenstein, 2012*),

we used the *C. parallelus* hybrid zone to investigate whether phage WO can also transfer between co-infections of B and F *Wolbachia*. Here, we present the first screen for phage WO in the *C. parallelus* hybrid zone. While we do not find evidence for WO transfer between B and F *Wolbachia*, we identify three main WO haplotypes in the grasshopper genome. We also report, for the first time to our knowledge, the transfer of large amounts of DNA from two divergent *Wolbachia* supergroups into the host nuclear genome.

## MATERIALS AND METHODS

### Sample collection, DNA extraction, and *Wolbachia* strain typing

The Spanish Comunidad de Madrid, the Gobierno de Aragón and the French Parc National des Pyrénées gave permission (permit numbers 10/103410.9/15; INAGA 500201/24/2012/12140; and Autorisation 2015-9, respectively) to collect *Chorthippus parallelus* individuals from five European and Iberian populations (Fig. 1). Gonads (or the whole body) were dissected and fixed in 100% ethanol. DNA was extracted as described elsewhere (*Martinez-Rodriguez, Hernandez-Perez & Bella, 2013*). *Wolbachia* was detected by PCR amplification of the *Wolbachia* 16S rRNA gene using *Wolbachia*-specific primers (*Zabal-Aguirre, Arroyo & Bella, 2010*), followed by nested PCR amplifications using B and F supergroup-specific primers (*Martinez-Rodriguez, Hernandez-Perez & Bella, 2013*). 10 µl of each amplification product were electrophoretically separated on 1% agarose gels, which were stained with 0.5 mg/ml ethidium bromide and visualized under UV light (UVIdoc, Uvitec Cambridge).

### Phage PCR amplification, cloning and sequencing

All PCR amplifications for phage and *Wolbachia* gene analyses were performed using 7.5 µl 2X GoTaq Green Master Mix (Promega), 3.6 µl sterile water, 1.2 µl of each primer (5 µM) and 1.5 µl template DNA for a 15 µl total reaction volume (scaled up as necessary) on a Veriti Thermal Cycler (Applied Biosystems) with the following primers: phgWOF (5′-CCCACATGAGCCAATGACGTCTG-3′) and phgWOR (5′-CGTTCGCTCTGCAAGTAACTCCATTAAAAC-3′) for the WO minor capsid gene (*Masui et al., 2001*); WolbF (5′-GAAGATAATGACGGTACTCAC-3′) and WolbR3 (5′-GTCACTGATCCCACTTTAAATAAC-3′) for the *Wolbachia* 16S ribosomal RNA gene (*Casiraghi et al., 2001*); ftsZunif (5′-GGYAARGGTGCRGCAGAAGA-3′) and ftsZunir (5′-ATCRATRCCAGTTGCAAG-3′) for *Wolbachia* ftsZ (*Lo et al., 2002*). The following primers were designed as part of this study to amplify specific WO alleles: forward primer WOPar1_F1 (5′-AATCTAAAAAGCGAAGTGAATCGTT-3′) paired with phgWOR to amplify Cpar-WO1 alleles; reverse primer WOPar3_R1 (5′-CGACAGTTCTCGTAGCCTTCCTCA-3′) paired with phgWOF to amplify Cpar-WO3 alleles.

To clone and sequence the *orf7* gene, PCR products were run on a 1% TBE agarose gel, then excised and purified using the Wizard PCR and Gel Clean-up Kit (Promega). 4 µl of each purified PCR product was cloned into a pCR4-TOPO vector using the TOPO TA Cloning kit (Invitrogen). OneShot TOP10 *E. coli* cells (Life Technologies)

were transformed with the recombinant plasmids through heat shock according to the manufacturer's protocol. Transformed *E. coli* were plated on LB + carbenicillin plates and incubated overnight at 37 °C. Fifteen to 26 colonies were picked per plate then sent to GENEWIZ, Inc. (South Plainfield, NJ) for plasmid purification and Sanger sequencing. Both forward and reverse directions were sequenced for each plasmid and then assembled in Geneious v5.5.8. For Sanger sequencing with allele-specific primers, PCR products were excised and purified from agarose gels as described above and then sent to GENEWIZ, Inc. for sequencing. Both forward and reverse directions were sequenced for each PCR product then assembled in Geneious v5.5.8.

## Phylogenetic tree construction

All multiple sequence alignments and phylogenetic trees were constructed in Geneious v5.5.8. Minor capsid sequences obtained through cloning and/or Sanger sequencing were aligned with homologous sequences from other WO phages (Table S3) using the Translation Align Tool with default parameters, and the *dnaA* and *fabG* contigs from high-throughput sequencing were aligned with their homologs in *Wolbachia* strains using the Geneious alignment tool with default parameters. *Wolbachia dnaA* and *fabG* genes were extracted from full genome sequences from NCBI (Genbank) as follows: *w*Ha (CP003884.1), *w*Mel (AE017196.1), *w*Ri (CP001391.1), *w*No (CP003883.1), *w*Pip strain Pel (AM999887.1), *w*Oo (HE660029.1), *w*Ov strain Cameroon (HG810405.1), *w*Bm strain TRS (AE017321.1), and *w*Cle (AP013028.1).

After indels were manually removed, the minor capsid gene alignment was 332 bp with 49 sequences, the *dnaA* alignment was 742 bp with 11 sequences, and the *fabG* alignment was 735 bp with 11 sequences. "N"s were added to the 5′ or 3′ ends of any sequences that were shorter than the total alignment length. jModelTest 0.1.1 was used to determine the best model of nucleotide evolution for each alignment based on the corrected Akaike information criterion (AICc). For each gene, PhyML (*Guindon & Gascuel, 2003*) and MrBayes (*Huelsenbeck & Ronquist, 2001*) were executed in Geneious with default parameters to construct a maximum likelihood tree with bootstrapping and a Bayesian tree with a burn-in of 100,000, respectively. For the minor capsid gene, the third best model of nucleotide evolution (HKY + G) was used to generate both the maximum likelihood and Bayesian trees since the first two best models were not available in PhyML or MrBayes. The Hasegawa–Kishino–Yano (HKY) model of nucleotide evolution allows variable base frequencies and separate rates for transitions and transversions (*Hasegawa, Kishino & Yano, 1985*). For the *dnaA* gene, the 10th best model of HKY + G was used since the first 9 were not available in PhyML or MrBayes. For the *fabG* gene, the second best model of GTR + G was used. The general time reversible (GTR) model of nucleotide evolution allows variable base frequencies and assumes a symmetric substitution matrix (*Lanave et al., 1984*; *Tavare, 1986*). For both the HKY and GTR models, rate variation among sites was modeled as a gamma distribution (+G).

## High throughput sequencing of *Wolbachia* genomic inserts

Pooled DNA from three uninfected grasshoppers (two gonadal and one whole-body extractions) from the Gabas population (pure Cpp) was sequenced as 100 bp, paired-end reads on a single lane of an Illumina HiSeq2000 at the Vanderbilt VANTAGE sequencing facility. All analysis of sequencing data was performed in CLC Genomics Workbench 8. Reads were trimmed based on a quality limit of 0.05 and minimum length of 50 bp. After trimming, the data consisted of 227,349,258 reads with an average length of 93.5 bp totaling 21,347,095,705 bp.

All reads were initially mapped to the B *Wolbachia* genome of *w*Pip strain Pel (Genbank AM999887) using the CLC mapping tool with the following parameters: 80% similarity over 80% read length, mismatch cost = 2, insertion cost = 3, deletion cost = 3, and random mapping of non-specific reads. To ensure that the mapped reads were indeed from *Wolbachia,* reads from core *Wolbachia* genes were searched against the NCBI nucleotide database using blastn (megablast). Since many of the reads were more similar to genomic sequences from the F *Wolbachia w*Cle than to *w*Pip or other B *Wolbachia* genomes, we re-mapped all reads to the *w*Cle (Genbank AP013028) and *w*Pip (Genbank AM999887) reference genomes simultaneously with more stringent parameters: 90% similarity over 90% read length, mismatch cost = 2, insertion cost = 3, deletion cost = 3, and random mapping of non-specific reads. Since read mapping to each genome was mutually exclusive, this generated a list of reads that preferentially mapped to one genome over the other. To ensure that this was the case, reads that mapped to *w*Pip were extracted and mapped to the *w*Cle genome and vice versa with the more stringent parameters (90% similarity over 90% read length) to generate a combined list of "non-specific reads". After excluding these non-specific reads, the remaining reads were mapped back to the genome that they preferentially mapped to in order to determine the final lengths of the B and F inserts.

To find genes shared between the inserts, we took the reads that preferentially mapped to either *w*Pip or *w*Cle (B and F reads, respectively) and mapped them with less stringent parameters (70% sequence similarity over 90% sequence length) to the reciprocal genome. Genes were considered shared between the two inserts if both B and F reads mapped to homologous genes on both the *w*Pip and *w*Cle genomes and total read length for both B and F reads on each gene exceeded 80 bp. B and F variants for each gene were manually verified by using blastn (discontiguous megablast) to confirm that percent similarity of B variants to *w*Pip were higher than to *w*Cle and vice versa.

To determine whether reads preferentially mapped to *w*Pip and *w*Cle over *Wolbachia* strains from other supergroups, we mapped all reads simultaneously to *w*Pip, *w*Cle, *w*Mel, *w*Bm, and *w*Oo reference genomes with a cutoff of 90% sequence similarity over 90% read length or 65% similarity over 80% read length. All reads that ambiguously mapped to more than one location were discarded.

Visualization of read mapping coverage on the *w*Pip and *w*Cle circular genomes was generated using the BLAST Ring Image Generator v0.95 (*Alikhan et al., 2011*) with a maximum mapping coverage of 30.

## FISH analysis

To perform the cytogenetic analyses, male adult specimens of Cpp and Cpe were collected from the Gabas (France) and Escarrilla (Spain) populations, respectively. Grasshopper gonads were extracted and fixed in fresh ethanol:acetic acid (3:1) and used to prepare slides. After identifying uninfected individuals with *Wolbachia*-specific primers, as mentioned above, we designed primers to amplify a *Wolbachia* contig (Cpar-Wb1) identified during genome sequencing (Table S6): 177contigF (5′-ACAGGAATTACAGCCTCAGGT-3′) and 177contigR (5′-AAAAGCGTGGCAACAAAGTT-3′). PCR amplifications used the following conditions: Buffer 1X, MgCl$_2$ 2 mM, dNTPs (Roche) 0.2 mM, 1.2 μM of each primer, BIOTAQ DNA polymerase 1.25 U (Biotools), and 100 ng of genomic DNA, adjusting the final volume to 25 μl. The PCR program started with a cycle of 3 min at 95 °C, followed by 35 cycles of denaturing (30 s at 95 °C), annealing (45 s at 56 °C), extension (3 min at 72 °C), and a final extension of 10 min at 72 °C. PCR products were run on a 0.7% TAE agarose gel and were purified using the Illustra GFX PCR DNA and Gel Band Purification kit (GE Healthcare).

The purified DNA from the PCR was used to generate FISH probes with the DecaLabel DNA Labeling kit (Thermo Scientific), which is based on the random-primed method (*Feinberg & Vogelstein, 1983*; *Feinberg & Vogelstein, 1984*), including a digoxigenin-labeled nucleotide. The complete reaction consisted of: 10 μl of decanucleotide, 5X Reaction Buffer, 1 μg of cDNA, and nuclease-free H$_2$O till 42 μl, keeping this mix at 100 °C for 10 min; afterwards, we added 1 mM dNTPs mix, 1.75 μl of Digoxigenin-11-dUTP (Roche), and 1 μl of Klenow enzyme then incubated at 30 °C for 2 h. Finally, the probes were purified again with the Illustra GFX PCR DNA and Gel Band Purification kit (GE Healthcare), and eluted in 50 μl of H$_2$O.

Chromosome slides were prepared from fixed gonads to observe hybridization to male meiotic chromosomes from Cpe and Cpp individuals. Gonads were adhered to slides by the conventional technique of squashing, and the coverslip was removed after immersing the slides in liquid nitrogen. The squashed biological material was then treated for 5 min with pepsin (50 μg/ml in 0.01 N HCl) at 37 °C, followed by a 30 min incubation in 2% paraformaldehyde at room temperature. Endogenous peroxidases were inactivated by incubation for 30 min with 1% H$_2$O$_2$. Slides were then dehydrated in a series of ethanol washes (70%, 85%, and 100%) and dried out. Slides were denatured and hybridized in the presence of 50 μl of the hybridization mixture under a coverslip for 5 min at 70 °C. Hybridization mixture was composed of 2 μl of labeled probe, 50% formamide, 2X SSC, 300 mM NaCl, 30 mM sodium citrate, pH 7.0. After denaturing, slides were left overnight in a wet chamber at 37 °C. Posthybridization washing and visualization of FISH-TSA (tyramide signal amplification) probes were performed as described previously (*Krylov, Tlapakova & Macha, 2007*; *Krylov et al., 2008*). Detection of probes with antidigoxigenin conjugated to horseradish peroxidase (Roche) was done at a concentration of 1:2,000 in TNB (Tris-NaCl-blocking buffer). The tyramide solution (Perkin Elmer) was incubated onto the slides for 5 min at a concentration of 1:50. Chromosomes were counterstained with 50 ng/μl of DAPI (4′,6-diamidino-2-phenylindole, Roche) diluted in Vectashield

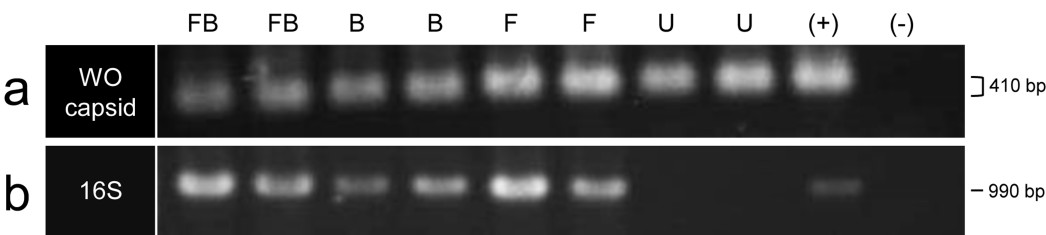

**Figure 2 PCR amplification of the (A) WO minor capsid (orf7) gene and (B) 16S ribosomal RNA gene.** Two individuals of each infection type are shown: FB, co-infected; B, B infection only; F, F infection only; U, uninfected; (+), positive DNA control; (−), no template negative control. For the WO capsid gene, the gel ran askew, making some bands appear larger in size than others though all bands represent the same sized PCR amplicon (410 bp).

(Vector Laboratories). Results were observed in a digital image analysis platform based on Leica DMLB fluorescence microscope with independent green and blue filters. Images were captured as tiff files using a cooled CCD Leica DF35 monochrome camera (Leica Microsystem), and final images were processed employing Photoshop CS6 (Adobe).

# RESULTS

## Infected and uninfected grasshoppers across the hybrid zone harbor phage WO genes

To initially determine the prevalence of phage WO in the *C. parallelus* hybrid zone, we PCR-screened hybrid, Cpe, and Cpp grasshoppers of all infection types (co-infected, B-infected, F-infected and uninfected, Table S1) for the minor capsid gene (*orf7*), a virion structural gene commonly used to identify WO haplotypes (*Bordenstein & Wernegreen, 2004*; *Chafee et al., 2010*; *Gavotte et al., 2004*; *Masui et al., 2000*). Surprisingly, *orf7* amplicons were detected in 42 out of 43 (98%) samples, including all uninfected grasshoppers ($n = 8$, Fig. 2A), which were determined to be *Wolbachia*-free using nested PCR for the *Wolbachia* 16S ribosomal RNA gene (Fig. 2B). Blank controls were negative for the *orf7* amplicon. These results indicate that (i) phage WO is or once was ubiquitous in *C. parallelus* and (ii) at least part of phage WO has laterally transferred to the grasshopper genome.

## Diverse WO haplotypes are present in the grasshopper genome

To identify phage WO variation in a hybrid zone population, we cloned and sequenced an approximately 350 bp region of *orf7* from a co-infected (604FB), B-infected (603B), F-infected (607F) and uninfected (641U) hybrid grasshopper from the Portalet population (Table S2). To confirm that these alleles were present in other individuals within the same population, we used allele-specific primers to amplify and sequence *orf7* from five additional individuals: three uninfected (167U, 169U and 186U), one F-infected (180F) and one co-infected (192FB). In total, we identified eight unique *orf7* alleles throughout the phylogenetic tree of select WO minor capsid sequences (Fig. 3, Table S3). Seven of these alleles clustered into three haplotypes (Cpar-WO1, Cpar-WO2, and Cpar-WO3) based on a 96% identity cutoff (Figs. 3 and 4). Since all three haplotypes

# WO Minor Capsid (*orf7*)

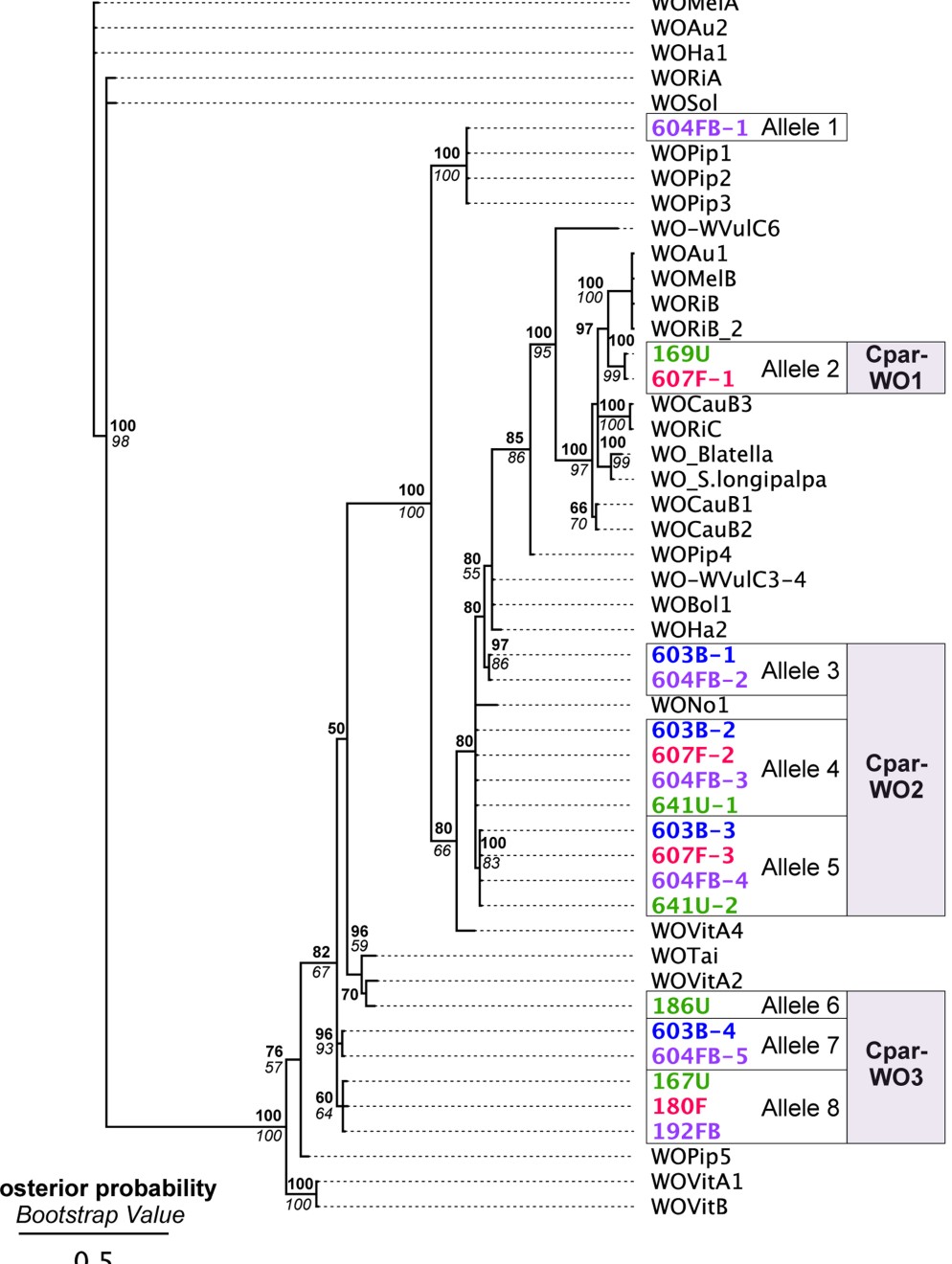

**Figure 3 Phylogeny of the WO minor capsid (*orf7*) gene.** Bayesian phylogeny constructed using indel-free nucleotide alignment of the phage WO *orf7* gene. Sequences generated in this study are labeled with individual identification numbers and color-coded based on the grasshopper's infection status: FB, co-infected (purple), B, B-infection only (blue), F, F-infection only (red) and U, uninfected (green). Numbers after a hyphen designate different *orf7* sequences from the same individual. Posterior probability (Bayesian) and bootstrap (maximum likelihood) values over 50 are indicated in bold and italics, respectively. Accession numbers for sequences used in the tree, including the sequences from this study, are listed in Table S3. The tree is arbitrarily rooted.

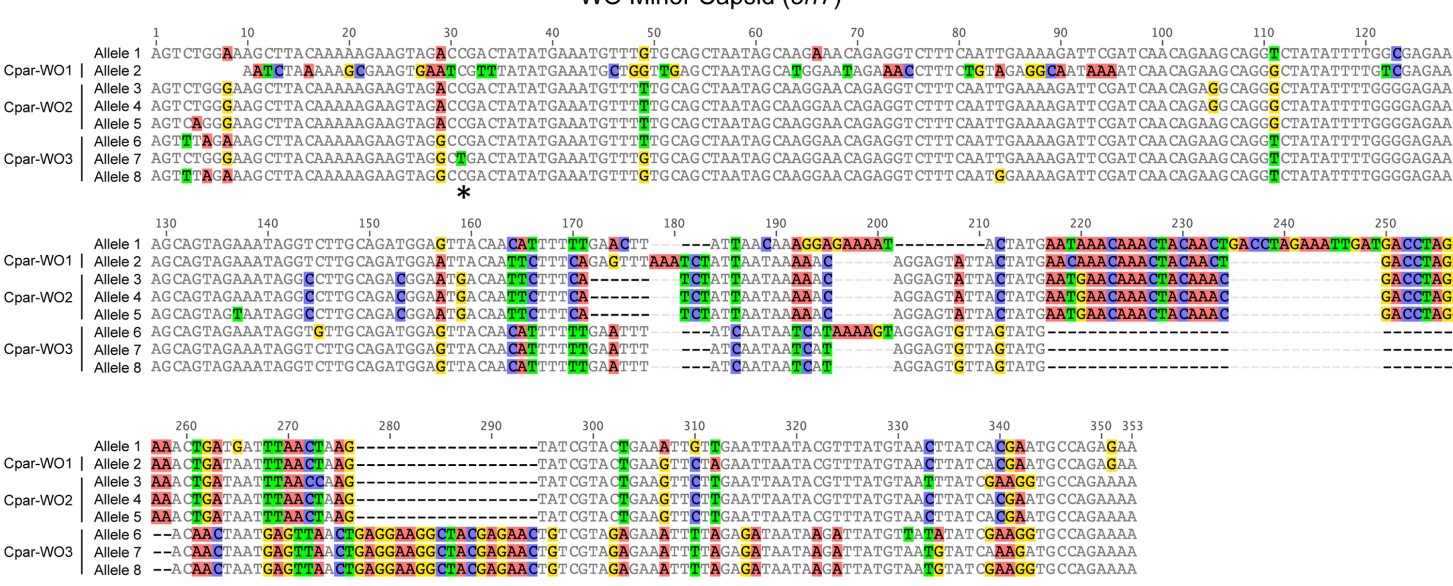

**Figure 4** **Nucleotide alignment of WO minor capsid (*orf7*) alleles from hybrid grasshoppers.** Asterisk indicates location of C to T substitution that introduces a premature stop codon in Cpar-WO3, allele 7. Nucleotides are counted from the start of the sequence alignment, not from the transcription start site of the gene.

contain sequences obtained from uninfected individuals, we conclude that at least three phage WO insertions are present in the grasshopper nuclear genome. Two alleles without an identical sequence from an uninfected individual (alleles 3 and 7) may actually be present in a cytoplasmic *Wolbachia* strain rather than the host genome, but we have conservatively clustered them within the Cpar-WO2 and Cpar-WO3 haplotypes, respectively, since they are each 97.7% identical to an allele from an uninfected individual (alleles 4 and 8, respectively). An additional *orf7* allele (allele 1) was only found in a single co-infected individual, so we cannot conclude whether it was sequenced from a cytoplasmic *Wolbachia* infection or a nuclear insert.

All alleles appear to be coding except for allele 7, which has a C to T substitution at nucleotide 31 that introduces a premature stop codon (Fig. 4). Since an identical allele was identified in another individual (604FB-5), it is unlikely that the SNP is a result of a PCR or sequencing error. Thus, at least one of the phage WO haplotypes may be undergoing pseudogenization, which is common for *Wolbachia* inserts in host genomes (*Brelsford et al., 2014*; *Nikoh et al., 2008*).

### Genome sequencing reveals B and F *Wolbachia* DNA inserts in the grasshopper genome

The unexpected finding of intact phage WO genes in uninfected grasshoppers led us to characterize the genomic inserts in the *C. parallelus* genome. To do so, we pooled DNA from three uninfected grasshoppers from the Gabas population, which is a pure Cpp population in the northern tip of the hybrid zone (Fig. 1). Cpp grasshoppers were chosen for sequencing instead of hybrid individuals to limit the amount of genetic variation in

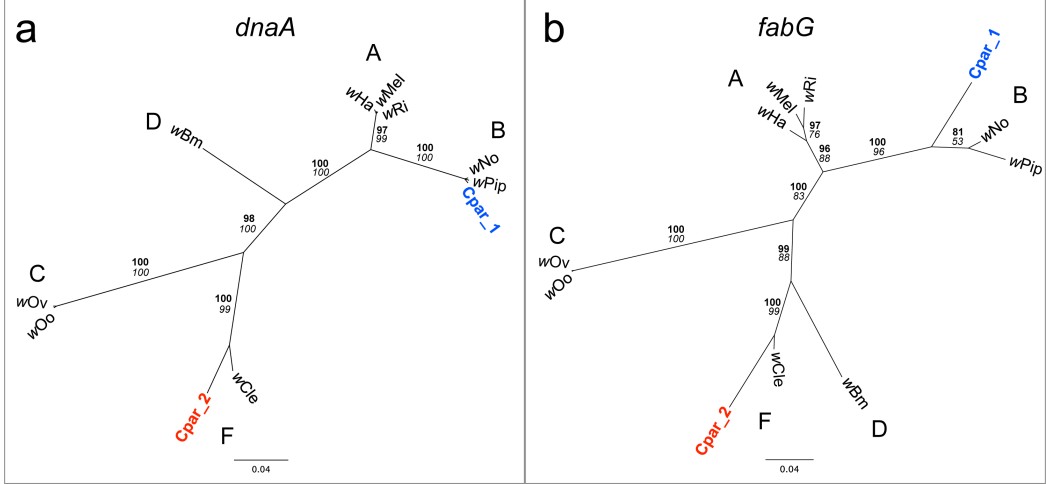

**Figure 5  Phylogenies of *Wolbachia dnaA* and *fabG* genes with *C. parallelus* genomic inserts.** Unrooted Bayesian phylogenies constructed using indel-free nucleotide alignments of *Wolbachia* (A) *dnaA* and (B) *fabG* genes with homologous contigs from *C. parallelus* genomic inserts (blue and red labels). *Wolbachia* supergroups (A–D, F) are indicated next to their respective clades. Posterior probability (bold) and bootstrap (italicized) values over 50 are indicated at each branch. Sequences for *dnaA* and *fabG* genes were extracted from the full genome sequences of their respective *Wolbachia* from NCBI (Genbank) as follows: *w*Ha (CP003884.1), *w*Mel (AE017196.1), *w*Ri (CP001391.1), *w*No (CP003883.1), *w*Pip strain Pel (AM999887.1), *w*Oo (HE660029.1), *w*Ov strain Cameroon (HG810405.1), *w*Bm strain TRS (AE017321.1), and *w*Cle (AP013028.1).

the sequencing and because the Gabas population has a high prevalence of uninfected individuals (*Zabal-Aguirre, Arroyo & Bella, 2010*). We used Illumina high-throughput sequencing to generate 227,349,248 paired-end reads with an average length of 93.5 bp after trimming. To extract WO reads from grasshopper sequences, we first mapped all trimmed reads with a cutoff of 80% similarity over 80% read length to the reference genome of the B *Wolbachia* strain *w*Pip from *Culex quinquefasciatus* mosquitoes (Pel strain, Genbank AM999887), which has five WO prophages (*Klasson et al., 2008*). However, in addition to phage-related reads, we found that many of the 22,833 reads that mapped to *w*Pip fell outside of the WO prophage regions. Altogether, phage and non-phage *Wolbachia* reads covered a total of 655,940 bp (44%) of the *w*Pip reference genome when non-specific reads (i.e., reads with more than one match to the reference genome) were allowed to map randomly.

Manual observation of SNPs across the alignment revealed that many of the genes appeared to have multiple alleles, some of which were more closely related to homologs in the genome of F *Wolbachia* strain *w*Cle (Genbank AP013028) than to those in the *w*Pip B *Wolbachia* strain. Indeed, phylogenetic analyses of small contigs containing portions of the *dnaA* (Fig. 5A) or *fabG* (Fig. 5B) genes show one contig grouping with *w*Cle and the other contig grouping with its homologs from strains *w*Pip and *w*No (both B *Wolbachia* strains).

To see if the sequencing reads preferentially map to *Wolbachia* from supergroups other than B or F, we simultaneously mapped all reads to the *w*Pip, *w*Cle, *w*Mel, *w*Bm, and *w*Oo reference genomes at a cutoff of 90% sequence similarity over 90% of read length. Reads

were only allowed to map exclusively to one genome, and reads that mapped ambiguously to more than one genomic location were discarded. In total, 84.7% of all mapped reads (14,424 out of 17,031) mapped preferentially to *w*Pip and *w*Cle (Table S4). A substantial number of reads (2,517) totaling 74,612 bp of the reference length also mapped to the genome of *w*Mel from the A supergroup. However, 63% of the *w*Mel reference covered by reads (47,054 out of 74,612 bp) are annotated as mobile genetic elements like phage WO, phage-associated regions adjacent to WO and transposases. Since phage WO and other mobile elements often transfer between *Wolbachia* strains (*Bordenstein & Wernegreen, 2004*; *Chafee et al., 2010*; *Gavotte et al., 2007*; *Masui et al., 2000*), these phage-related reads in the grasshopper genome do not necessarily originate from an A *Wolbachia* genome. Furthermore, the average contig length of those contigs mapping outside of the phage regions is only 113.4 bp (N50 = 100 bp), while contigs that map to phage and mobile elements average 229.5 bp (N50 = 321 bp). With such short contigs in the non-phage regions, any mutational drift in the inserts due to relaxed selection could cause reads to incorrectly map to a supergroup that differed from that of the original donor.

When mapping parameters were relaxed to 65% similarity over 80% read length, the number of reads mapping to all five genomes increased considerably, although the top two genomes with the most mapped reads and longest length of reference sequence covered were still *w*Pip and *w*Cle (Table S4). Again, a substantial number of reads mapped to *w*Mel but those contigs in the non-phage regions only averaged 75 bp in length (N50 = 50 bp), while those in phage regions were 228.6 bp long on average (N50 = 438 bp). Likewise, contigs mapping to *w*Bm (D supergroup) and *w*Oo (C supergroup) only averaged 55 bp and 47 bp, respectively. With the longest and thus most reliable contigs mapping to *w*Pip, *w*Cle, or phage regions, we conclude that most, if not all, *Wolbachia*-related reads in the grasshopper genome likely transferred from either a B or F *Wolbachia* strain.

When all trimmed reads were mapped simultaneously to only the *w*Pip and the *w*Cle genomes with cutoffs of 90% sequence similarity over 90% of read length and non-specific reads mapping randomly, 14,030 reads covering 493,855 bp and 3,768 reads covering 166,490 bp mapped to the *w*Pip and *w*Cle genomes, respectively (Fig. 6). Together, both mappings covered a total of 660,345 bp, which is similar to the 655,940 bp covered when mapping to *w*Pip alone at an 80% sequence similarity over 80% read length cutoff, supporting the hypothesis that *Wolbachia* DNA in the grasshopper genome originated from both the B and F supergroups. To verify that reads mapped preferentially to one supergroup over the other, reads that mapped to either *w*Pip or *w*Cle were reciprocally mapped to the other genome with the same parameters as before (90% sequence similarity over 90% of read length). Only 12.5% of reads that mapped to *w*Pip also mapped to *w*Cle, while 18.6% of reads that mapped to *w*Cle also mapped to *w*Pip. This means that, in total, 89.1% of reads (15,332 out of 17,798) preferentially mapped to one supergroup over the other. After removing the non-specific reads, the reads that preferentially mapped to each genome covered approximately 448 kb of the *w*Pip and 144 kb of the *w*Cle reference genomes. We note appropriate caution that this analysis does not allow us to distinguish whether these are large, intact inserts or multiple smaller inserts spread throughout the genome.

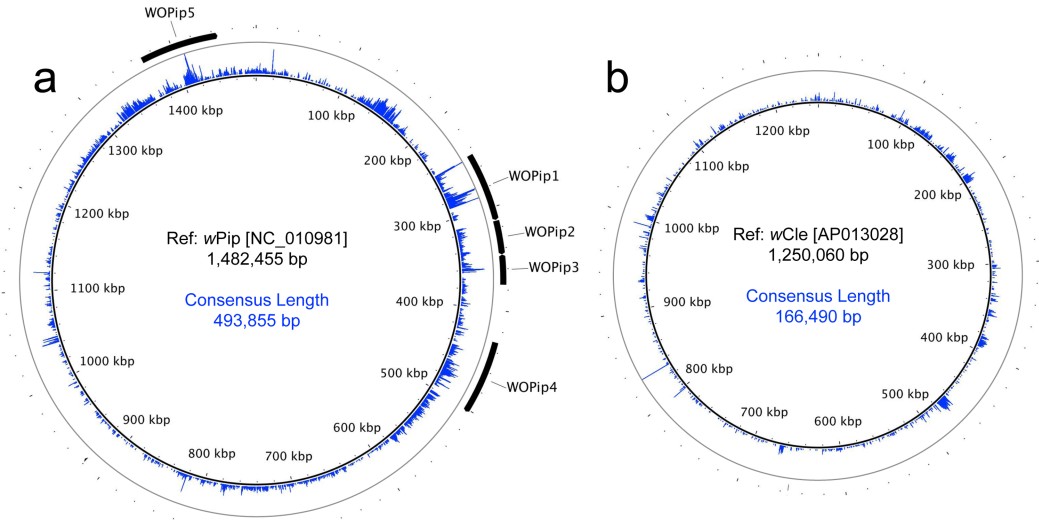

**Figure 6 Circular maps of sequencing coverage across the reference genomes of (A) *w*Pip and (B) *w*Cle.** Mapping coverage at each base is represented in blue on the inner rings with the max coverage set at 30 (outer gray circles). WO phage regions are indicated with black arrows.

To further analyze the dual origin of the *Wolbachia* gene transfers, we computationally searched for evidence of B and F *Wolbachia* inserts that contain similar genetic repertoires. In particular, we sought homologs in which the *w*Pip and the *w*Cle reference genes were both covered by B- and F-specific reads of at least 80 bp. We then used blastn to verify that reads for each gene homolog from one insert had a greater percent sequence similarity to *w*Pip than to *w*Cle and vice versa. In total, we found 130 homologous genes that met these criteria (Table S5), supporting a dual origin of the inserts.

### Genome sequencing confirms multiple WO haplotypes in the grasshopper genome

Given the diversity of *orf7* alleles sequenced from uninfected hybrid grasshoppers, it is not surprising that when read coverage was mapped onto the *w*Pip (Fig. 6A) and *w*Cle (Fig. 6B) reference genomes, areas of higher coverage clustered mostly in the prophage regions (Fig. 6A). After extracting and assembling contigs from reads that mapped to the five WO minor capsid (*orf7*) genes in *w*Pip, we confirmed that there are at least three *orf7* alleles in the uninfected Cpp grasshopper genome (Fig. S1). One allele (WO2-contig) is 97.3% identical to allele 4 from the Cpar-WO2 haplotype (Fig. S1). The other two alleles are most similar to sequences from the Cpar-WO3 haplotype: WO3-contig1 is 97.5% identical to allele 6 and WO3-contig2 is 100% identical to allele 7 (Fig. S1). We did not find any *orf7* alleles from the Cpar-WO1 haplotype in the genomic contigs, which may be a consequence of low sequencing coverage. However, if Cpar-WO1 is absent from the Cpp genome, then it may be specific to the Cpe subspecies or could even be unique to hybrids if the horizontal transfer occurred after establishment of the hybrid zone.

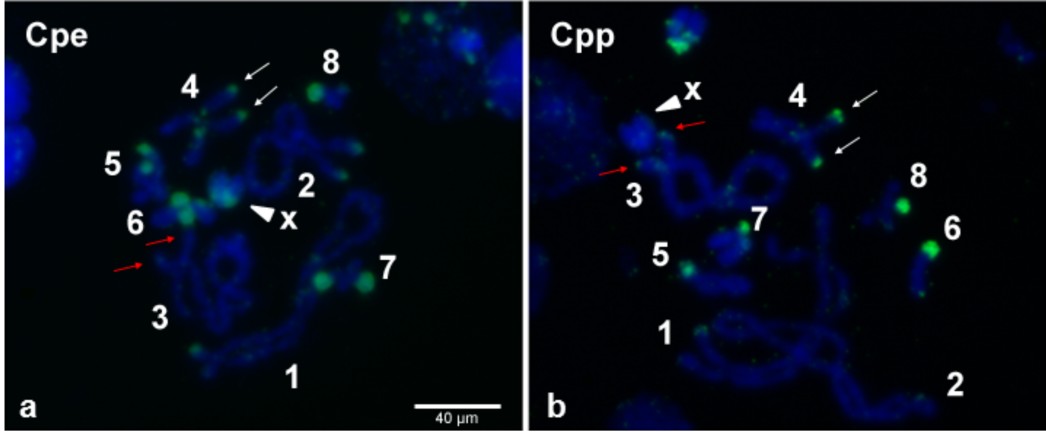

**Figure 7** ***Wolbachia* inserts localized to *C. parallelus* chromosomes.** Tyramide-coupled FISH Fluorescein signals using the Cpar-Wb1 probe reveal presence of *Wolbachia* genomic inserts (green fluorescence) in *C. parallelus erythropus*, Cpe (A) and *C. parallelus parallelus*, Cpp (B) meiotic chromosomes (blue fluorescence). Hybridization of *Wolbachia* insertions is abundant in telomeric regions of several chromosomes, certain interstitial regions and on chromosome X (arrowhead). White arrows mark a *Wolbachia* insert that coincides in homologous chromosomes of both Cpe and Cpp, while red arrows indicate a subspecies-specific insert present in Cpp but not Cpe. Numbers correspond to chromosome pairs (bivalents). Scale bar = 40 µm.

### FISH localizes *Wolbachia* inserts in grasshopper chromosomes

Even though, on average, 70% of individual grasshoppers from the Gabas population are uninfected with *Wolbachia* (*Zabal-Aguirre, Arroyo & Bella, 2010*), it is possible that the "uninfected" grasshoppers from Gabas had a low-titer *Wolbachia* infection that accounts for the sequencing of copious *Wolbachia* genes. This explanation is highly unlikely because PCR for two essential bacterial genes, 16S rRNA and *ftsZ*, failed to detect a product in all three grasshoppers pooled for sequencing, while PCR of WO genes amplified a band in all individuals for the *orf7* gene (Fig. S2). Moreover, to confirm *Wolbachia* insertions in the grasshopper genome, we used tyramide-coupled FISH to physically map *Wolbachia* genomic insertions in Cpe (Fig. 7A) and Cpp (Fig. 7B) chromosomes of uninfected male individuals. Hybridization of fluorescent DNA probes designed from a contig from the B *Wolbachia* insert (Table S6) revealed a discrete, repeatable distribution pattern along chromosomes in the karyotype (Fig. 7), particularly in telomeric constitutive heterochromatin and in some interstitial regions. When comparing the distribution of this contig on the chromosomes of Cpp and Cpe, some signals are present at homologous chromosomal locations in both genomes, such as on chromosome 4 (Fig. 7, white arrows), while other inserts, like that on chromosome 3 in Cpp (Fig. 7, red arrows), are subspecies-specific, suggesting that the former are ancestral to the last common ancestor of Cpp and Cpe, whilst the latter appeared after taxon divergence.

### DISCUSSION

The *Chorthippus parallelus* hybrid zone is an excellent model for symbiosis research since *Wolbachia* infection status is highly variable, with individuals collected at the same

geographical location infected with either F or B *Wolbachia*, co-infected with both or naturally uninfected (*Zabal-Aguirre, Arroyo & Bella, 2010*). Though *Wolbachia* diversity has previously been investigated in this system, this work comprises the first screen for *Wolbachia's* temperate phage WO. We set out to characterize the types of phage WO present in the population and to determine whether co-infection with *Wolbachia* strains from divergent B and F supergroups facilitated transfer of phage WO between *Wolbachia*. Instead, we discovered an unexpected diversity of phage WO *orf7* alleles and multiple instances of horizontal transfer of the phage WO *orf7* gene to hybrid and non-hybrid grasshopper genomes. In total, we identified eight unique *orf7* alleles from nine different individuals collected from a single hybrid population. Genome sequencing of Cpp grasshoppers confirmed that three of these alleles (4, 6, and 7) predate secondary hybridization of Cpp and Cpe subspecies, while the other alleles may have been introduced to the hybrid zone by Cpe or may be unique to hybrid populations.

Since many of the alleles are so similar to others ($\geq$96% identical), they may represent allelic variation at the same locus in the diploid grasshopper genome instead of independent gene transfers. Thus, we conservatively classified similar alleles into three phage "haplotypes". Interestingly, we did not conclusively identify *orf7* alleles that were specific to *Wolbachia* cytoplasmic infections even though many of the grasshoppers were infected by B and/or F *Wolbachia*. It is likely that the cytoplasmic *Wolbachia* infections harbor phage WO with *orf7* sequences that are so similar to those in the host genome that we cannot distinguish between the two. For example, alleles 3 and 7 were only sequenced from B-infected individuals and may reside in the cytoplasmic B *Wolbachia* genome, but further genome sequencing of the cytoplasmic *Wolbachia* is needed to verify this observation.

After sequencing the genome of uninfected Cpp grasshopers, we discovered that not only phage genes had transferred to the host genome but also large regions of both B and F *Wolbachia*. Many animal hosts that harbor or once harbored *Wolbachia* have evidence of *Wolbachia* DNA in their genomes (*Bordenstein, 2007*; *Dunning Hotopp, 2011*; *Dunning Hotopp et al., 2007*), probably because *Wolbachia* are uniquely poised for symbiont-to-host gene exchange since they target the germ-line stem cell niche during host oogenesis (*Fast et al., 2011*; *Robinson, Sieber & Dunning Hotopp, 2013*; *Toomey et al., 2013*). *Wolbachia* nuclear inserts can be quite large and cover a substantial portion of a *Wolbachia* genome. For example, approximately 30% of a *Wolbachia* genome is inserted in the X-chromosome of the bean beetle *Callosobruchus chinensis* (*Kondo et al., 2002*; *Nikoh et al., 2008*), while an estimated 180 kb of *Wolbachia* DNA is present in the genome of the longicorn beetle *Monochamus alternatus* (*Aikawa et al., 2009*). Multiple *Wolbachia* insertions in the same host genome have also been identified. Several *Drosophila ananassae* populations have multiple copies of an entire *Wolbachia* genome on one of their chromosomes (*Dunning Hotopp et al., 2007*; *Klasson et al., 2014*), while the tsetse fly *Glossina morsitans morsitans* genome has three *Wolbachia* chromosomal inserts with the two largest inserts each covering roughly half a *Wolbachia* genome at 527 kb and 484 kb (*Brelsfoard et al., 2014*). The large *Wolbachia* inserts in this case are highly similar to each other and also closely-related to the tsetse fly cytoplasmic *Wolbachia* strain, *w*Gmm, suggesting a single transfer from

*w*Gmm to the tsetse fly genome followed by duplication of the insert, though independent transfer events cannot be ruled out (*Brelsfoard et al., 2014*). Either way, both insertions came from the same *Wolbachia* supergroup and likely from the same *Wolbachia* strain.

In our study, phylogenetic analyses of variable contigs mapping to the same *Wolbachia* genes revealed that inserts in the *C. parallelus* genome likely originated from both B and F *Wolbachia*. To our knowledge, this is the first case of substantial *Wolbachia* DNA transfer from divergent supergroups into the same host genome. Similar techniques used to analyze the genomes of *Wolbachia*-free nematodes such as *Acanthocheilonema viteae, Onchocerca flexuosa, Loa loa,* and *Dictyocaulus viviparus* found ancient remnants of *Wolbachia* genes that appear to have originated from multiple supergroups when compared to present-day cytoplasmic *Wolbachia* genes (*Desjardins et al., 2013*; *Koutsovoulos et al., 2014*; *McNulty et al., 2010*). However, the antiquity of these horizontal transfer events makes accurate phylogenetic inferences difficult, especially since the *Wolbachia* genes in the nematode host are no longer under the same selective pressures as cytoplasmic *Wolbachia* genes. For example, *McNulty et al. (2010)* estimates that "fossilized" evidence of *Wolbachia* sequences in the genomes of *A. viteae* and *O. flexuosa* must be several million years old based on their low percent identities (78% and 81%, respectively) to any contemporary *Wolbachia* sequences. In contrast, average percent identities of the B *Wolbachia* gene variants to *w*Pip and the F *Wolbachia* gene variants to *w*Cle for the 130 shared genes in the *C. parallelus* inserts (Table S5) are 94 ± 0.05% and 93 ± 0.04%, respectively.

The higher percent identity to a contemporary *Wolbachia* strain for the grasshopper inserts suggest that they have transferred more recently and/or are better preserved in the grasshopper genome due to the unique evolutionary dynamics of grasshopper genomes. Orthopterans like grasshoppers, locusts and crickets are known for their enormous genomes, and *C. parallelus* grasshoppers have one of the largest genomes in the order with estimates ranging from 12.3 to 14.7 Gb (*Lechner et al., 2013*). Genome gigantism in Orthoptera is thought to largely be due to frequent acquisition of new genetic material coupled with slow rates of DNA loss (*Bensasson et al., 2001*; *Bensasson, Zhang & Hewitt, 2000*; *Song, Moulton & Whiting, 2014*). For example, Orthopteran genomes exhibit unusually high rates of DNA transfer from mitochondria to the nuclear genome (*Bensasson, Zhang & Hewitt, 2000*; *Song, Moulton & Whiting, 2014*), and, with the slow rate of DNA loss, some of these inserts have remained intact for 150 million years (*Song, Moulton & Whiting, 2014*). Based on the rate of mitochondrial gene acquisition, grasshopper genomes may be more amenable to horizontal gene transfer in general, especially from intracellular cytoplasmic entities like mitochondria or *Wolbachia*. It is not surprising, then, to presume that DNA from both B and F *Wolbachia* would eventually wind up in the *C. parallelus* host genome.

The dynamic nature of *Wolbachia* lateral gene transfer to the *C. parallelus* genome is evident when visualized with FISH. Some inserts are present at the same position on the chromosomes of both Cpp and Cpe while other inserts are subspecies-specific, indicating that insertion events likely occurred both before and after the divergence of the subspecies. Our sequencing of the WO *orf7* gene supports this hypothesis since the Cpar-WO2 and

Cpar-WO3 haplotypes are present in the genomes of Cpp individuals from Gabas and in hybrids from Portalet, while the Cpar-WO1 haplotype was only detected in Portalet. Subspecies-specific sequences are likely relatively young since the two subspecies are estimated to have diverged between 0.2 and 2 MYA (*Cooper & Hewitt, 1993*; *Lunt, Ibrahim & Hewitt, 1998*). If hybrid-specific inserts arose independently, they would be even younger since the transfer would have had to occur after the formation of the hybrid zone roughly 9,000 years ago (*Hewitt, 1993*; *Shuker et al., 2005a*). Thus, slow rates of DNA loss coupled with relatively recent transfer events allows standard phylogenetic analyses to easily identify and distinguish the inserts in the *C. parallelus* genome as originating from either a B or F *Wolbachia*, whereas *Wolbachia* inserts in nematode genomes may be too divergent to accurately predict the donor *Wolbachia's* supergroup.

Instead of independent transfers, B and F *Wolbachia* strains may have recombined to produce a *Wolbachia* strain with genes from both supergroups and part of this "hybrid" *Wolbachia* genome transferred as a single event into the *C. parallelus* genome. This scenario appears unlikely as we identified 130 *Wolbachia* genes with multiple alleles from both B and F *Wolbachia* in the genomic inserts. A recombinogenic genome with substantial genetic redundancy of essential genes is improbable given that endosymbiont genomes tend to be relatively streamlined (*Newton & Bordenstein, 2011*; *Wernegreen, 2002*). Furthermore, FISH analyses verified the presence of *Wolbachia* DNA in multiple locations on the *C. parallelus* chromosomes and further characterization of the inserts and their evolutionary history is in progress.

## CONCLUSION

Alongside genetic introgression, animal hybrid zones offer an avenue for symbiont exchange, especially for heritable endosymbionts like *Wolbachia* (*Mandel, Ross & Harrison, 2001*; *Zabal-Aguirre, Arroyo & Bella, 2010*). Resulting co-infections of multiple *Wolbachia* strains in a hybrid host provide opportunities for genetic exchange within the intracellular arena (*Bordenstein & Reznikoff, 2005*; *Metcalf & Bordenstein, 2012*; *Newton & Bordenstein, 2011*). Though exchange of bacteriophage WO occurs often between co-infections of A and B *Wolbachia* (*Bordenstein & Wernegreen, 2004*; *Chafee et al., 2010*; *Kent et al., 2011*; *Masui et al., 2000*), we found no evidence for phage WO transfer among B and F *Wolbachia* in hybrid *C. parallelus* grasshoppers. Instead, we found that horizontal gene transfer is clearly a dynamic process in *C. parallelus*, with two discrete *Wolbachia* supergroups (B and F) transferring approximately 448 kb and 144 kb of DNA, respectively, to the host genome. Since many insects are co-infected with *Wolbachia* from different supergroups, it is curious why there are not more insect genomes with *Wolbachia* inserts of dual origin. Part of the answer is likely that other genomes with inserts of dual origin have simply not been sequenced yet. However, grasshopper and other Orthopteran genomes, with their high rates of DNA acquisition and slow rates of DNA loss, may be uniquely poised for acquiring *Wolbachia* genes and maintaining them relatively intact for long periods of time, allowing phylogenetic analyses to accurately distinguish between different supergroups. Though the gigantic genomes of Orthopterans currently make them challenging to sequence and

assemble, it will be interesting to see if more species of this undersampled insect order also have DNA from multiple endosymbionts in their genomes.

## ACKNOWLEDGEMENTS

We thank Sarah Bordenstein for advice on the project. We are grateful to the Spanish Comunidad de Madrid, the Gobierno de Aragón and the French Parc National des Pyrénées for permission to collect grasshoppers. We thank Prof. JM Szymura for orientating us to the Tyramide-coupled FISH to detect small DNA sequences and to Prof. V Krylov for his advice to set up this technique.

### Funding

This work was supported by the National Institutes of Health (grant number RO1 GM085163 to SRB, and CBMS training grant T32 GM 008554 to LJF) and the National Science Foundation (grant number DEB 1046149 and IOS 1456778 to SRB, and Graduate Research Fellowship 0909667 to LJF). The Spanish work was supported by the Spanish MINECO (I+D+i grant CGL2012-35007 to JLB.) and the collaboration of Chromacell S.L. The funders had no role in study design, data collection and analysis, decision to publish, or preparation of the manuscript.

### Grant Disclosures

The following grant information was disclosed by the authors:
National Institutes of Health: RO1 GM085163.
CBMS training: T32 GM 008554.
National Science Foundation: DEB 1046149, IOS 1456778.
Graduate Research Fellowship: 0909667.
Spaish MINECO: CGL2012-35007.
Collaboration of Chromacell S.L..

### Competing Interests

The authors declare there are no competing interests.

### Author Contributions

- Lisa J. Funkhouser-Jones and Miguel Pita conceived and designed the experiments, performed the experiments, analyzed the data, wrote the paper, prepared figures and/or tables, reviewed drafts of the paper.
- Stephanie R. Sehnert performed the experiments, analyzed the data, reviewed drafts of the paper.
- Paloma Martínez-Rodríguez performed the experiments, wrote the paper, reviewed drafts of the paper.
- Raquel Toribio-Fernández performed the experiments, reviewed drafts of the paper.

- José L. Bella and Seth R. Bordenstein conceived and designed the experiments, analyzed the data, contributed reagents/materials/analysis tools, wrote the paper, reviewed drafts of the paper.

## Field Study Permissions

The following information was supplied relating to field study approvals (i.e., approving body and any reference numbers):

The Spanish Comunidad de Madrid, the Gobierno de Aragón and the French Parc National des Pyrénées gave permission (10/103410.9/15; INAGA 500201/24/2012/12140; and Autorisation 2015-9, respectively) to collect Chorthippus parallelus individuals from five European and Iberian populations.

## DNA Deposition

The following information was supplied regarding the deposition of DNA sequences:

GenBank KR081342–KR081347

GenBank KT599860–KT599861

Sequence Read Archives SAMN03469681.

## Supplemental Information

Supplemental information for this article can be found online at http://dx.doi.org/10.7717/peerj.1479#supplemental-information.

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
