# Peer review of "Wolbachia co-infection in a hybrid zone: discovery of horizontal gene transfers from two Wolbachia supergroups into an animal genome"

_PeerJ, doi:10.7717/peerj.1479_

## Round 0.1 · original submission · Minor Revisions

Please address the comments raised by reviewer 1, in particular regarding clarification of the relative ages of transfers in grasshoppers and nematode genomes, and why you may argue that evidence for diverse strain origins is stronger in the case of grasshopper genomes. Please also address the Reviewer comments regarding mapping of Wolbachia reads to genomes other than wPip (supergroup B) or wCle (supergroup F) and provide some information regarding the nature of the largest regions of the Wolbachia genome identified as transfers into the grasshopper genome.

Reviewer 1 ·

Basic reporting

No comments

Experimental design

No comments

Validity of the findings

No comments

Additional comments

The submission by Funkhouser-Jones and colleagues is a well-written manuscript that reports an interesting study of Wolbachia co-infections in a hybrid zone and documents substantial ancient lateral gene transfer (LGT) from disparate Wolbachia endosymbionts into the host grasshopper genome. I enjoyed reading it

My only major comment concerns the text between Lines 281 and 294. The authors correctly note (Lines 273 – 280), that Wolbachia DNA fragments in Wolbachia-free nematodes appear to originate from multiple Wolbachia supergroups and that the antiquity of these LGTs coupled with little / no selection pressure likely contributes to this phenomenon. The authors then develop an argument (lines 281 – 294) that attempts to prove that the situation in grasshoppers is different, but I’m not totally convinced. The authors mention that “inserts in C. parallelus, on the other hand, are of much more recent origins (discussed below)…” but the ensuing text does not provide any information on the age of the transfers and is thus rather weak in this regard. They simply state that orthopteran genomes are large, perhaps due to acquisition of new DNA coupled with a slow rate of DNA loss, and that some DNA transferred to the nuclear genome from mitochondria has remained nearly intact over millions of years. This text does not inform on the age of the LGTs they report. Furthermore, if acquired DNA in orthopterans is lost slowly and can remain intact over millions of years then aging of LGTs in grasshoppers is likely to be very different to aging LGTs in other organisms. In nematodes for example, an LGT detection threshold of 80% similarity over 80% of the read length (as used in the initial identification of LGTs in this study) would allow identification of highly degenerated transfers that would be more likely to map to diverse Wolbachia supergroups. But if the threshold in nematodes is made more stringent then the number of LGTs found is reduced, but the more recent / better preserved sequences are obtained. LGTs that differ at just one or two nucleotide positions compared to their Wolbachia endosymbiont can be found in species that still contain endosymbiont. It would be interesting to know how LGTs in grasshoppers map to the various Wolbachia supergroups if a more relaxed threshold (perhaps 65% similarity) is used.

Minor comments:

Line 46: “alphaprotobacteria” should be alphaprotobacrerium
Lines123 – 138 and Fig 3, Table S2: I found this section difficult to follow. The problem is that initially only 4 grasshoppers (with unique identifiers) were examined. These revealed 6 unique orf7 alleles (as correctly written), but Fig. 3 indicates 8 alleles. It took me time to realize that Fig 3 also includes alleles found in a second analysis that included additional individual grasshoppers. Although the manuscript and figure are fine once the reader has realized this, it was not immediately clear to me, and so the wording in the text or Figure legend should be improved to make it easier to grasp this point. A second confusion arises in Line 128 where 7 unique alleles are discussed and then in Line 136 an 8th unique allele is introduced. For clarity, I suggest that the sentence on line 128 begins “In total, we identified 8 unique alleles, 7 of which clustered ……” and the sentence beginning on Line 136 begins “The unique orf7 allele (allele 1)….”
Line 173: I think it important to remind the reader here that the 655940 bp covered when mapping to wPip alone was done using the more relaxed 80% similarity over 80% of the read length. Otherwise the logic behind the presented numbers is not clear.
Lines 181 – 183: can the authors comment on the largest contiguous regions of wCle and wPip that were completely covered by reads?
Line 210: I think the term “ruled out” is too strong and would prefer “appears highly implausible” or something similar. If the Zabal-Aguirre reference cited elsewhere in the manuscript convincingly showed that uninfected grasshoppers were present in this geographic area then it would be useful to include it again here to strengthen the authors argument that a low titer Wolbachia infection is unlikely.
Line 244: delete second “S”: from “Disscussion”
Line 231: word “of” would be better as “with”
Line 243: “grasshopper” should be plural
Line 274: correct spelling is “viteae”
Lines 275 - 280: please look carefully at the cited Koutsovoulos reference. Radopholus is mentioned briefly in this paper, but the main focus is on Dictyocaulus viviparous, a non-filarial nematode with remnants of Wolbachia in its genome. I think this species is intended instead of R. similis?
Line 281: insert “other” after “the”
Line 343: primers for which supergroups were used? All or just B and F?
Line 367: Need symbol for centigrade and new sentence shouldn’t start with a numerical (“15-26…).
Line 383 and throughout manuscript: the usual designation for the Wolbachis from O. volvulus is wOv in keeping with more other nematode Wolbachia.
Lines 376 – 397: any non-default parameters for software used would be helpful. Otherwise state “with default parameters”
Lines 393 – 397: can the authors describe the acronyms HKY+G more fully or provide a suitable reference or both?
Lines 409 – 415; the authors mention that many reads appeared to map preferentially to wCle rather than wPip. They should provide information on how many reads mapped preferentially to Wolbachia genomes other than Wpip (supergroup B) or wCle (supergroup F). this information is important in consideration of my main comment (raised above).
Line 457: “over” seems an odd word choice. Is ‘ to observe hybridization to male…” better?
Line 458: “developed” is inappropriate here as the squashing part is not part of slide development
Lines 468, 469, 470: TSA, POD & TNB should be defined

·

Basic reporting

The paper investigates the possible transfer of the bacteriophage WO in Wolbachia strains that cross a hybrid zone of meadow grasshoppers (Chorthippus parallehus) in Europe. Interestingly, the study was set up to determine if the WO bacteriophage was transferred across two subspecies (C. parallelus erythropus-Cpe and C. parallelus parallelus-Cpp in their respective Wolbachia parasites (B and F), but discovered that the WO genome (at least 3 haplotypes) were found in the grasshopper genome. In addition, there is a large amount of Wolbachia DNA also found in the host nuclear genome, which is the first report of this type of transfer.

Experimental design

The experimental design is sound, and although the initial questions that promoted the research did not show any transfer, the results that were obtained were interesting and provide a different aspect on host-symbiont interactions.

Validity of the findings

The manuscript is well written, and provides some interesting and unexpected results on horizontal transfer of both phage WO and Wolbachia DNA into the host (grasshopper) genome. The results give credence to earlier infection of WO to the Cpp populations prior to their separation and subsequent secondary hybridization. Interestingly, the Wolbachia gene transfer evidence provides additional information regarding horizontal gene transfer between symbionts and hosts. This is particularly relevant- given that Wolbachia target germ-line cells in oogenesis, a perfect place to insert their DNA into the future generations of hosts (and has been shown in a number of arthropod hosts). What makes this work unique is that both B and F genotypes were found to be transferred in the Cp genome. I find this part of the paper quite fascinating, and will lead to an new avenue of research for this hybrid zone of grasshoppers.

---

## Round 0.2 · accepted · Accept

The revisions of the manuscript have extensively and adequately addressed the issues raised by the reviewers and have provided important new information that strengthens some of the main points put forward in this work.